# Quantifying atmospheric and land drivers of hot temperature extremes through explainable Artificial Intelligence

Arnau Garcia  ${\sf Mesa^1}$  , Lluís  ${\sf Palma^1}$  , Markus  ${\sf Donat^{1,2}}$  , Stefano Materia  $^{1,3}$  , and Raül Marcos Matamoros  $^4$ 

Correspondence: Arnau Garcia Mesa<sup>1</sup> (arnau.garcia@bsc.es)

**Abstract.** Different drivers have been shown to play a central role in modulating the occurrence and intensity of summer temperature extremes, yet their individual contributions remain difficult to quantify. In this study, we develop an explainable machine-learning framework to disentangle the respective influences of large-scale atmospheric circulation, soil-moisture anomalies, and rising CO<sub>2</sub> concentrations on boreal-summer temperature extremes at six locations across Europe and North Africa with different characteristics of land-atmosphere coupling (Córdoba, Lyon, Hannover, Stockholm, Belgrade, and Marrakech). Using SHapley Additive exPlanation (SHAP) values, we find that the atmospheric circulation consistently dominates model explainability across all locations, contributing to 67-90 % of the total mean SHAP, with the geopotential at 500 hPa field contributing the most. Soil-moisture influence exhibits a northward gradient: negligible at Marrakech (0.5 %), moderate at Córdoba (7.7 %), and substantial at Lyon (15 %). Additionally, negative correlations between soil-moisture standardized anomalies and SHAP values across three depth levels corroborate the amplifying effect of land drying on heat extremes. We demonstrate the robustness of these findings to a less stringent (80th percentile) extreme definition. Furthermore, the identified driver contributions are consistent when using alternative observational data for temperature extreme definition and for computing SPI/SPEI drought indices as proxies for soil moisture, with SPEI showing a closer alignment to the original ERA5-Land results. We also illustrate the methodology for case studies of two individual events, heatwaves occurring in Córdoba (Spain) 2021 and Hannover (Germany) 2018, which reveal a pronounced spatial pattern in the distribution of SHAP values for the circulation predictors. They also confirm the enhanced role of the land component in regions of Northern Europe and reveal a contribution of the anthropogenic factor through CO2 concentrations, even for specific events. These insights enhance our understanding of the physical mechanisms behind temperature extremes and demonstrate the potential of explainable artificial intelligence methods to quantify the contributions from different drivers of hot temperature extremes.

<sup>&</sup>lt;sup>1</sup>Earth Sciences Department, Barcelona Supercomputing Center (BSC), Barcelona, Spain

<sup>&</sup>lt;sup>2</sup>Institució Catalana de Recerca i Estudis Avançats (ICREA), Barcelona, Spain

<sup>&</sup>lt;sup>3</sup>CMCC Foundation–Euro–Mediterranean Center on Climate Change, Bologna, Italy

<sup>&</sup>lt;sup>4</sup>Facultat de Física, Universitat de Barcelona, Diagonal 645, 08028 Barcelona, Spain

#### 20 1 Introduction

Understanding weather extremes is of vital importance due to their impacts on society and ecosystems. Events such as extreme precipitation, severe droughts, and intense heatwaves pose growing threats to biodiversity and sectors like agriculture, energy, health or the economy (e.g., Zscheischler et al., 2018). As the climate warms, evidence from observations and model simulations shows that these events are becoming longer, more intense and more frequent (Intergovernmental Panel On Climate Change, 2023). Consequently, predicting, characterizing, and understanding these events is crucial for the development of early-warning systems and adaptation strategies in a climate change context.

Much progress has been made in identifying the physical processes driving temperature extremes, although many mechanisms are still not well understood due to difficulties in quantifying their interactions and responses to climate change (Barriopedro et al., 2023). The changing behaviour linked to the positive trends in global temperatures poses challenges for establishing a unique definition or common metrics describing hot extremes (Perkins and Alexander, 2013).

Hot extremes are primarily driven by a combination of atmospheric, ocean, and land surface conditions, including persistent high-pressure systems and soil moisture deficits (Barriopedro et al., 2023). In this study, we discard the ocean drivers because we are focusing on the short term variability drivers of hot extremes. Additionally, we include atmospheric CO<sub>2</sub> concentrations to capture the multiyear trend of hot extreme events. Many studies focus in understanding the role of these land–atmosphere interactions, such as the positive soil-moisture–temperature feedback, and its amplification role during high-pressure conditions (Materia et al., 2022; Quesada et al., 2012). This amplification becomes especially relevant in a changing climate due to the evidence of regional shifts of wet, dry and transitional evaporative regimes (Hsu and Dirmeyer, 2023). Seneviratne et al. (2006), for instance, highlight the need of including land–atmosphere interactions in regional simulations to correctly modulate the increase in summer temperature variability (and resulting temperature extremes) in central and Eastern Europe due to increasing greenhouse gas concentrations.

Regarding the role of the atmospheric circulation, many studies show the direct link of hot extremes to atmospheric circulation patterns, such as persistent double jets over Eurasia (Rousi et al., 2022), mid-tropospheric deep depressions over the North–Atlantic (D'Andrea et al., 2024), or episodes of Saharan warm air intrusion in the West Mediterranean (Cos et al., 2024).

In addition, growing attention is being paid to heatwave-driving mechanisms related to other elements of the Earth system,

like particular patterns in preceding sea surface temperatures identified as precursors of summer heatwaves in Europe (Beobide-Arsuaga et al., 2023).

Complementing physical understanding, new tools and methodologies are being actively explored to improve the identification and prediction of temperature extremes. In particular, Artificial Intelligence (AI) and Machine Learning (ML) approaches are emerging as powerful complements to traditional numerical modelling techniques (Camps-Valls et al., 2025). Moreover, ongoing developments in explainable AI (XAI) offer promising pathways to interpret model behaviour, allowing researchers to assess the contribution of individual drivers to extreme events (Materia et al., 2024). This capability is critical not only for attribution purposes, but also for potentially gaining new insights into the underlying physical mechanisms.

https://doi.org/10.5194/egusphere-2025-5392 Preprint. Discussion started: 19 November 2025

© Author(s) 2025. CC BY 4.0 License.

For example, Toms et al. (2020) demonstrate how explainable ML methods such as layerwise relevance propagation (LRP) and backward optimization can be used to interpret neural network predictions in climate applications, revealing scientifically meaningful relationships between input patterns and network outputs. Davenport and Diffenbaugh (2021) employs LRP to analyse large-scale circulation patterns associated with extreme precipitation. Other explainable AI (XAI) techniques such as SHAP have also been used to assess the relative importance of different climate drivers; for instance, Uckan et al. (2024) apply SHAP values within a Random Forest framework to quantify the contribution of multiple heatwave drivers at the global scale.

In this study, we aim to disentangle the respective roles of atmospheric circulation, soil moisture and anthropogenic climate change in driving extreme temperature events during the boreal summer (JJA), using an explainable ML modelling framework. With this goal in mind, we aim to develop a framework that is both predictive and diagnostic.

First, we classify daily extreme events using the 90th percentile threshold of the climatological maximum temperature distribution. Then, we build a robust extreme/no-extreme binary classifier using supervised ML, with lagged anomalies of large-scale circulation variables and soil moisture, and atmospheric CO<sub>2</sub> concentrations as inputs. To understand the drivers learned by the ML model during the prediction task, we apply the SHAP explainability technique to quantify the contribution of each input feature to the model's predictions (Lundberg and Lee, 2017). First, we compute general aggregated SHAP values for each variable to obtain a general characterization of feature relevance. Then, we proceed by analysing specific SHAP explanations for two case studies to investigate region-specific driver behaviour: the August 2021 heatwave in Córdoba and the July 2018 heatwave in Hannover. These events were selected to represent contrasting soil-moisture feedback regimes – arid versus temperate – with differing soil moisture variability and land-atmosphere coupling strengths (Hsu and Dirmeyer, 2023). We extend our explainable ML analysis to evaluate the robustness of our driver attribution to different extreme thresholds (80th and 95th percentiles) and to the use of alternative observational data for the extreme event classification and the land component features.

In section 2 we discuss the data and methodology, including the datasets used, the domain for circulation data, the selected locations based on different soil-moisture feedback regimes, the deep learning model used and the explainability method. In sect. 3 we show and discuss the results: we display the extreme classification results, give a statistical assessment of the model, and provide the SHAP explainability results. Lastly, conclusions of the study can be found in sect. ??.

## 2 Data and methodology

#### 2.1 Datasets

We use data from ERA5 (Soci et al., 2024), the fifth-generation atmospheric reanalysis from the European Center for Medium-Range Weather Forecasts (ECMWF), which provides a globally complete, gridded data from 1940 to the present. The long-term availability and spatial completeness of reanalyses make them suitable for studying climate trends, variability, or extreme events despite substantial uncertainties across datasets (Donat et al., 2014). We use daily data using the spatial domain discussed in 2.3. We use atmospheric circulation fields that represent the state of the atmosphere and constitute the large-scale variables

in the study: Geopotential at 500 hPa (g500) [m<sup>2</sup>s<sup>-2</sup>], Geopotential at 200 hPa (g500) [m<sup>2</sup>s<sup>-2</sup>], and Sea Level Pressure (psl) [hPa].

For land-surface variables, we use the ERA5-Land dataset, which has a better representation of land-surface variables due to its 9km horizontal resolution (Muñoz-Sabater et al., 2021). The variables we use are daily maximum 2-meter temperature (TX) to define our target events, and soil moisture at three depth levels (*swvl1*, *swvl2*, and *swvl3*) as predictors. TX from ERA5-Land is only used to define hot extremes, as the prediction target of the ML model.

Lastly, we include seasonal atmospheric CO<sub>2</sub> concentrations to account for the trend of climate change. We employ U.S. National Oceanic and Atmospheric Administration (NOAA) measurements collected at the Mauna Loa Observatory in Hawaii. The concentrations used have a constant value for each summer and are a seasonal mean.

Circulation and land variables are used as lagged standardized anomalies (explained in 2.4). For each prediction day, the model is trained on features derived from values recorded on preceding days. For the local-scale, soil moisture is averaged over the previous seven days to account for its relatively low daily variability (Miloshevich et al., 2023). The values of circulation variables are taken from the day before the prediction, corresponding to a one-day lag in atmospheric features. The inclusion of up to three lags for the circulation features was tested, but we decided to keep a simpler model because the accuracy did not improve by including more lags (not shown).

Similar variable sets using atmospheric circulation and land features have been used in Koh et al. (2025), where they developed a machine-learning framework combining spatial extreme-value theory and adapted loss functions to study large-scale heat extremes, or in Miloshevich et al. (2023), who predict the probability of occurrence of long-lasting 14-day heatwaves over France.

## 2.2 Observational data and drought indices

To test the robustness of our results to the data source, we supplement the analysis based on reanalysis with observational data for the daily maximum temperature and the land component. For the atmospheric circulation features, we keep the data from ERA5. For the local-scale daily maximum temperature used in the extreme event definition, we use directly the E-OBS gridded dataset (version 29.0) of TX (Cornes et al., 2018).

Due to the lack of long-term, consistent observational soil moisture records, we employ the Standardized Precipitation Index (SPI) (McKee et al., 1993) and the Standardized Precipitation Evapotranspiration Index (SPEI) (Vicente-Serrano et al., 2010) as proxies for soil moisture anomalies, computed with the E-OBS data. These indicators have already been assessed as surrogates of in situ and satellite soil moisture observations (Scaini et al., 2015; Wang et al., 2015; Yuan et al., 2020), with generally larger association when potential evapotranspiration is included. We use the Hargreaves method (Masmoudi-Charfi and Habaieb, 2014) for computing the evapotranspiration, essential for the SPEI calculation, and the gamma distribution to standardize the data. Precipitation and temperature data are sourced from the E-OBS dataset. As for soil moisture, both SPI and SPEI grid-points are averaged spatially over a 100kmx100km region.

The timescale for the index computation was selected by computing the Spearman correlation of the drought indices with the three soil moisture levels of the original model (see figures C1 and C2). Soil moisture is a variable with a memory that

extends beyond a few days (Vicente-Serrano et al., 2010), so the drought index must be computed on a scale that reflects this persistence. Based on the correlation analysis, the selected timescale for the SPEI and SPI indices is 30 days (SPEI30d and SPI30d). This timescale shows the best correlation of the indices with *swvl1* and *swvl2*. which we show to be in general the most relevant depth levels in our explainability analysis (see 3). We also computed the correlation using a 7-day rolling mean of the soil moisture data—the same preprocessing applied before inputting soil moisture to the model—and obtained very similar results, with correlation values not differing more than 0.07. Additionally, we computed the drought indices using ERA5 data to validate the correlations computed with the E-OBS gridded data, with correlation between datasets not differing more than 0.1. In the correlation analysis, we did not lower the timescale of the drought indices computation (e.g. 15 days) to stay within the original monthly definition of the indices (Vicente-Serrano et al., 2010), and to avoid fitting the normalized distribution to data that may not be sufficiently smooth.

## 2.3 Atmosphere data domain and target locations

With the atmospheric pressure and geopotential height fields, we aim to capture the general circulation influencing the locations where we predict extreme temperature events. The large-scale domain must be large enough to cover the area of interest, but not too large to avoid including unrelated atmospheric variability (Mahlstein et al., 2015). Fig. 1 shows the selected domain marked as a red box. The full North Atlantic Ocean is included, since studies link this region to the development of extreme temperature events in various European regions (Bischof et al., 2023). A large part of North Africa is included as well to capture warm and dry intrusions that may enhance the likelihood of extremes (Cos et al., 2024). The northern and eastern limits are extended enough to include synoptic-scale circulation patterns.

Figure 1. Large-scale data domain and locations selected for the local-scale data in the prediction target. The red box indicates the domain used in the study for the ERA5 data, with longitudes ranging from 54° W to 70° E and latitudes from 14° N to 71° N. Locations are illustrative for different climate regimes in Europe.

https://doi.org/10.5194/egusphere-2025-5392 Preprint. Discussion started: 19 November 2025

© Author(s) 2025. CC BY 4.0 License.

140

150

155

We study the roles of circulation, soil moisture and anthropogenic climate change during extreme temperature events in six regions around Córdoba, Marrakech, Lyon, Belgrade, Hannover, and Stockholm (see fig. 1). For each site, local-scale data is averaged over a surrounding 100 km × 100 km region, to smooth out unwanted variations across space. The chosen locations span different evaporative regimes: from water-limited (Marrakech) to energy-limited (Stockholm). Location choice is also partly based on literature: Córdoba and Marrakech are studied in Lemus-Canovas et al. (2024), Stockholm's 2018 heatwave in Wilcke et al. (2020), and Iberia and Central Europe heatwaves in Khodayar Pardo and Paredes-Fortuny (2024). Belgrade was selected to provide a more eastern perspective, potentially influenced by sea surface temperatures and atmospheric anomalies in other regions discussed in Beobide-Arsuaga et al. (2023).

Lastly, the robustness analysis using observational data (discussed in 2.2) could not be carried out for Marrakech, due to missing data for several years (see Fig. 4).

## 2.4 Climatology, anomalies and the extreme temperature definition

Both the climatology and 90th percentile are computed only for the reference period 1950–2000, within the model's training period (1950–2014). This avoids an indirect provision of future information to the model, which would leak evaluation data into training. The training period was extended to 2014 to include a period with a clear trend in the yearly count of extremes (see fig.4).

First, we calculate the raw climatology by taking the daily average. Then, as in Mahlstein et al. (2015), we apply a local polynomial regression (LOESS) to obtain a smoothed profile. After a few tests with the window timescale, we selected a 30-day smoothing window for both climatology and standard deviation. Anomalies are computed daily by subtracting the smoothed climatology from the original data, and then standardized using the standard deviation. Subtracting the climatology removes the annual cycle, and applying the LOESS smoothing removes the short-term variability. Extended detail on LOESS is given in appendix A1.

Using the extreme definition in Perkins and Alexander (2013) an extreme event is defined when the daily mean maximum temperature exceeds a specified percentile of the climatological distribution. Following Beobide-Arsuaga et al. (2023), the percentile is calculated over a centered 5-day window using the reference period, and then we apply LOESS smoothing with a 30-day window. The primary analysis in this study uses the 90th percentile, but to test the robustness of our results to the extreme event definition, we conduct additional analyses using the 80th and 95th percentiles, and all are computed following the same procedure.

# 2.5 The deep learning model

Although we use daily data since 1950, the number of extreme days based on the 90th percentile remains low (see fig.2). This imbalance is one of the central challenges we need to cope with during the model training, to prevent a default to the non-extreme class. On the other hand, the limited number of extreme days increases the risk of overfitting in our binary classification task when using many parameters, something that needs to be taken into account when defining the architecture and training. The imbalance problem is even worse if one uses higher percentiles in the extreme definition (see Table B1).

**Figure 2. Climatology and percentiles for the region of Córdoba.** Day of the year (x-axis) and daily maximum 2-meter temperature from ERA5-Land (y-axis). We show the raw climatology and percentiles, together with the 5-day window pull and LOESS smoothing versions. The text box shows the total frequency of heat wave events, and the separated frequencies for the reference period 1950–2000, and the rest of the time series (2001–2024).

The main contributor to high parameter dimensionality are the large-scale fields, which include spatial dimensions at  $0.25^{\circ}$  resolution, unlike local-scale data that is area-averaged and provides time-only vectors. Flattening large-scale data would result in  $n_{var} \cdot n_{lags} \cdot n_{lat} \cdot n_{lon}$  inputs, making the model overly complex. We first reduce dimensionality by re-gridding large-scale data to  $1^{\circ}$ , directly reducing computational cost and overfitting risk. We think this coarser resolution still captures atmospheric circulation features for extreme temperatures.

We use a model architecture designed to capture land–atmosphere coupling (see fig. 3). It includes a Multi Layer Perceptron (MLP) for land and CO<sub>2</sub> data, and a ConvNeXt convolutional neural network (CNN) (Liu et al., 2022) for spatial features from circulation, which, to our knowledge, it has not been used before in driver quantification studies. Their outputs are concatenated and passed to a final MLP for binary classification of extreme or non-extreme on a daily basis. This architecture is hereafter referred to as the *CombinedModel*.

The MLP for the Land State + CO2 inputs has two hidden layers of 15 and eight nodes, inspired by Mayer and Barnes (2021), and the MLP for the final classification has three hidden layers with 64, 32 and 8 nodes. The architecture of the ConvNeXt model is explained in detail in Liu et al. (2022). ReLU is used as activation function in the MLPs, and softmax applied at the output to yield probabilities (Goodfellow et al., 2016). A 0.5 threshold determines the predicted class, kept constant throughout the study. Additionally, we use a 20-member ensemble to filter out noise and handle part of the epistemic uncertainty of the model. These ensembles are generated by changing the initial random seed (detailed in 2.7).

**Figure 3. Diagram of the Combined Model.** Includes a simple MLP for the Land-State plus a ConvNeXt model for the Atmosphere state, which are then combined using a final MLP that does the binary classification. An ensemble of 20 members is used during training to add robustness and treat the model's epistemic uncertainty.

The optimizer used for the CombinedModel is the AdamW optimizer (Loshchilov and Hutter, 2019). We train the model with an 80 % and 20 % random split of the samples for the training and validation respectively, and apply a random shuffle. The Optuna Python package is used for the tuning of the following hyperparameters of the ML model: batch size, learning rate, weight decay and class weights of the predicted classes. The model is trained for 75 epochs unless validation loss increases for a total of five epochs. The score used in the hyperparameter tuning process is the balanced accuracy minus the final validation loss, which is the best validation loss found before early-stopping.

#### 2.6 Explainability of the machine learning model using SHapley Additive exPlanation values

To quantify the contribution of each input feature to the model's predictions, we use SHapley Additive exPlanations (SHAP), a game-theoretic approach that explains the output of any machine learning model by attributing it to the input features (Lundberg and Lee, 2017).




Positive SHAP values mean a positive influence on the likelihood of the predicted class by the model, while a negative SHAP value means that the feature decreases the likelihood of predicting the class. The magnitude of the value quantifies the influence of the feature, and SHAP values are zero for irrelevant. SHAP values are an additive feature attribution method, meaning that the contribution of each feature can be computed independently, and all values summed up.

The sum of the SHAP values across all features equals the model output (prediction) minus the expected (baseline) prediction :

$$f(x) = \phi_0 + \sum_{i=1}^{N} \phi_i, \tag{1}$$

where f(x) is the model prediction for the input x,  $\phi_0$  is the baseline (computed by averaging the model prediction over training data),  $\phi_i$  is the SHAP value for the feature i and N is the total number of features in the model. The additive nature of SHAP values allows us to compute a value per grid-point, highlighting the regions the model finds most important, and to spatially aggregate these values into global importance or combine them across features, as done in 3.2.

We compute SHAP values in the testing period using GradientExplainer, which is designed for differentiable models, such as deep neural networks, and estimates them by using an extension of the integrated gradients method (Sundararajan et al., 2017). We believe this choice to be appropriate given the evaluation metrics results for the Integrated Gradients method shown in Bommer et al. (2024), though no XAI method evaluation was carried for our study.

# 2.7 Creating robust explanations

Since our focus is to study the contribution of the atmospheric circulation, the soil moisture, and atmospheric CO<sub>2</sub> concentrations during extreme temperature events, we want to ensure that our explanations are done for predictions for which our model is confident. To do so, we take an ensemble of trained models with twenty members, adding robustness to our output probabilities and computed SHAP values. The members are created by setting deterministically twenty different seeds, initializing the random processes present in the framework. On top of that, we analyse the general explainability results for a certain percentage of top most confident predictions, following the approach of studies like Mayer and Barnes (2021).

We assess model performance using balanced accuracy (BA) and the Area Under the Receiver Operating Characteristic Curve (AUC), which are well-suited for imbalanced classification tasks.

#### 220 3 Results and discussion

Fig. 4 shows the yearly count of summer extremes (true labels) for the six selected locations, resulting from the extreme definition given in 2.4. We show the results of the extreme classification both for the ERA5-Land and E-OBS datasets. Clear warming trends are apparent for Córdoba and Lyon regions. Belgrade region also shows a visible trend after approximately 1960. The average count during summer for the period 1950–2000 is approximately 10 for all locations, which is expected considering our 90th percentile definition of extreme. When comparing the classification results for ERA5-Land and E-OBS


**Figure 4. Extremes yearly count for the six locations for both ERA5-Land and E-OBS data.** Yearly extreme count for the full period of used data in ERA5 and (1950–2024) E-OBS (1950–2023 for E-OBS). Years in the x-axis and count (number of extreme days) in the y-axis. The text box shows the mean count for the period 1950–2000 to verify the percentile definition of extreme in both datasets.

datasets, we see a general agreement except for some periods and locations like the 50s decade in Córdoba, or the years 1950 and 1952 in Belgrade.

#### 3.1 Prediction and statistical assessment of the models

For the primary analysis using the 90th percentile threshold, table 1 in ??, gives relevant statistics of the trained Combined-Model on the test data using the 50% most confident predictions. Fig. B1 shows how we assess the CombinedModel model robustness by plotting the evolution of accuracies when varying the confidence threshold. For each member individually, the CombinedModel shows a smooth increase in both balanced and extreme accuracy across all locations. The improvement in both accuracy measures with increasing network confidence is an expected behavior. For some locations, low percentages of confidence (see fig. B1) show a trade-off between extreme and non-extreme accuracies, with the former increasing and the latter decreasing, until a minimum is reached and non-extreme accuracy starts improving again. We did not explore this behaviour in depth, but it may be related to how we handle class imbalance.

In Fig. B2 we show the ROC curve change when changing the percentage of confidence. When using the ensemble of 20 members, most locations show a displacement towards area under the ROC curve equal to 1.

The model trained on the 80th percentile definition shows a similar statistical robustness for the accuracies and ROC curves

B3. The increased number of extreme events mitigates class imbalance, resulting in smooth and consistent improvements in

balanced accuracy and AUC values as the confidence threshold increases. This indicates the model is well-trained, and its predictions are reliable for this definition. In contrast, the model performance for the 95th percentile definition is notably weaker for most locations, with the balanced accuracy behaving poorly when increasing the percentage of confidence in the predictions. The drastically reduced number of extreme events (see table B1) presents a significant challenge for training, leading to less stable behavior in accuracy metrics and ROC curves as a function of prediction confidence (see Fig. B4). The class imbalance and lack of extreme events samples is a known issue in ML application for weather extremes (Miloshevich et al., 2023; Sha et al., 2025), so the results using the 95th percentile are not surprising. Due to this lack of robust predictive skill, we do not consider the explainability results for the 95th percentile definition to be trustworthy.

Finally, the model trained with observational TX and SPI/SPEI indices for the 90th percentile also demonstrates robust predictive performance, providing confidence in the reliability of its explanatory outputs (not shown). 250

Table 1. Statistical assessment of the CombinedModel for the different locations, considering the 50% most confident predictions. For the test data, we show the extreme class accuracy (EA), the non-extreme class accuracy (NEA), and the balanced accuracy (BA), which accounts for false positives and false negatives. TP, FP, TN, and FN stand for true positives, false positives, true negatives, and false negatives respectively. The last column shows the area under the ROC curve (AUC).

| Site   | EA[%]       | NEA[%]   | BA[%]    | TP  | FP | TN  | FN | AUC  |
|--------|-------------|----------|----------|-----|----|-----|----|------|
| Cór.   | 100±0       | 90±2     | 95±1     | 203 | 27 | 230 | 0  | 0.97 |
| Lyon   | $100 \pm 0$ | 91±2     | 96±1     | 207 | 23 | 230 | 0  | 0.96 |
| Hann.  | $100 \pm 0$ | $76\pm3$ | $88\pm1$ | 158 | 72 | 230 | 0  | 0.94 |
| Stock. | $100 \pm 0$ | $72\pm3$ | 86±1     | 140 | 90 | 230 | 0  | 0.90 |
| Bel.   | $100 \pm 0$ | 75±3     | 87±1     | 152 | 78 | 230 | 0  | 0.94 |
| Marra. | $100 \pm 0$ | $76\pm3$ | 88±2     | 159 | 71 | 230 | 0  | 0.95 |

## 3.2 SHAP values



Taking the 50% most confident extreme predictions, Fig. 5 shows the mean SHAP values across samples in the test dataset for each of the features in the model. We calculate a global SHAP value for the Land and Atmosphere compounds by summing the contributions of their respective features, which can be done thanks to SHAP values being an additive feature attribution method. To determine the SHAP percentage for a feature, we first calculate the mean of its absolute SHAP values over the entire time series. This time-averaged mean is then expressed as a percentage of the total mean SHAP value contributed by all features.

The atmospheric circulation has the largest importance (67-90%) at all locations. Apart from being intuitive, this result is in line with for instance Uckan et al. (2024), who show that geopotential height at 500 hPa is the dominant driver of hot extremes worldwide, particularly in mid-latitudes, and that the relevance of atmospheric drivers increases with the duration of the hot event. The Land SHAP value is more important in northern regions. In Marrakech, the land SHAP percentage is practically zero, which is reasonable given the region is generally very dry during the summer, therefore the interannual variability of soil


Figure 5. Extreme class prediction mean SHAP value percentage for all locations using the CombinedModel taking the 50% most confident predictions. In the x-axis we have the percentage with respect to the total sum of mean absolute SHAP values for all features, and in the y-axis the labels of the features used as inputs in the prediction. The text box shows the aggregated percentage for the Land and Atmosphere components, and the CO<sub>2</sub> SHAP percentage.

moisture is low and consequently coupling with atmosphere very little. Córdoba shows a higher percentage (7.7 %) explained by the land SHAP compared to Marrakech. The land SHAP value increases for higher-latitude locations, being specially important in Lyon. This result is in line with Jach et al. (2022), who assessed how changes in the mean temperature and moisture influence the land–atmosphere coupling strength over Europe.

As for the large-scale fields, there is a predominance of the geopotential at 500hPa, with only Marrakech having sea level pressure being more important than the atmospheric fields at higher altitude.

The atmospheric  $CO_2$  feature has a higher importance in regions that exhibit a stronger trend in extreme temperature days (see Fig. 4). Except for Stockholm and Hannover, the importance of the  $CO_2$  feature is generally high (>10%).

Lastly, in Fig. 6 we can observe the negative relationship between the standardized anomalies of the soil moisture levels and the contribution to the extreme class prediction. The results are consistent, with negative anomalies across levels contributing positively to predicting the extreme class and the other way around for positive anomalies. The figure also reflects the larger spread and importance for the first soil moisture level at this timescale, especially in northern regions. This key negative



Figure 6. SHAP values swvl[1,2,3] across samples test phase for the 50% most confident predictions and extreme class. The x-axis represents the SHAP value of the sample and the y-axis the soil moisture level feature used in the model. The colour bar shows the value of the standardized anomalies of the soil moisture level.

correlation is also present in the SHAP results for both the 80th percentile model (not shown) and the model using SPEI D3, further confirming the robustness of the identified land-drying feedback mechanism across different methodological choices.

The discussed explainability results are robust to the definition of an extreme event. For the 80th percentile model, the SHAP values (see Fig. D1) show the same main characteristics: the circulation component remains dominant (64.6–88.9%), and the land component still exhibits a northward gradient in importance, though less pronounced. Except for Córdoba and Marrakech, the relative contribution of soil moisture is, however, generally smaller than for the 90th percentile model. This would suggest a more marginal role of soil moisture in less severe heat events for which the land–atmosphere feedbacks, a key amplifier of extreme temperatures, may not be fully activated. The fact that the land component contribution is similar for Marrakech and Córdoba in the two extreme definitions might be an indication of the land–atmosphere coupling being rarely activated there in summer due to the scarce coupling in very dry conditions, independently of the extreme definition.

The driver contributions are also consistent when using observational data from E-OBS instead of ERA5-Land for TX and the SPEI drought index as a proxy for the soil moisture variables. In general, the XAI results reproduce the established north—south gradient in land driver importance and the dominant importance of the atmospheric circulation component (see Fig. D2). Córdoba is the only location that shows a notable discrepancy from the original model when using the swvl features, with the value being approximately four times smaller when calculated with the SPEI index. Nonetheless, the finding that Córdoba's value is smaller than those of more northern locations remains robust, and the small value is physically consistent considering the very dry conditions in summer for this location. In Fig. D3 we see that we also have a negative correlation



**Figure 7. SHAP values extreme class prediction for the Córdoba case study.** a): SHAP values temporal evolution for the circulation features and the three soil moisture levels. b): SHAP spatial distribution for the SHAP peak-day of *g500* lag 1. Statistical significance was assessed via a bootstrapping procedure. The 95 % confidence interval for the mean SHAP value was determined by using the 2.5th and 97.5th percentiles of the resulting bootstrapped distribution. c): Standardized anomalies of geopotential at 500hPa in colour and data in contours for the geopotential height at 500hPa for the predicted day (13 August 2021). The black dot in the panel *b* marks the location of Córdoba.

between SHAP value and SPEI30d feature values. When using the SPI index, the XAI results for the land component are less similar to the original ERA5-Land soil moisture results (see Fig. D4). Although a correlation analysis between the indices and soil moisture shows that SPEI generally correlated slightly better with soil moisture than SPI (see C1), the differences are relatively small. Nevertheless, the XAI outcomes reveal that SPI captures less of the original soil moisture signal. This aligns with physical understanding, as SPEI, by incorporating evapotranspiration, provides a more comprehensive representation of surface moisture balance and its coupling with the atmosphere than SPI, which is based solely on precipitation.

Overall, the feature with the largest importance shown by the XAI results for the g500 field. As a result, we decided to test the robustness of the model when using only this field to represent the atmospheric circulation features, obtaining similar results to the ones discussed above (see Fig. D5). The predominance of the g500 field is not surprising, as it is known to have a key role in determining the state and evolution of the troposphere, as well as a key indicator of the climate change response (Christidis and Stott, 2015). Lemus-Canovas et al. (2024) and Quesada et al. (2012) also show the important role of the atmospheric circulation component in triggering the heatwave event, and the land compound acting as an amplifier of extreme conditions. Moreover, Wilcke et al. (2020) highlight the importance of circulation in the 2018 summer heatwave over Stockholm, and its capability to force warm, long-lasting conditions without much contribution of surface feedbacks.


**Figure 8. SHAP values extreme class prediction for the Hannover case study.** a): SHAP values temporal evolution for the two circulation features and the three soil moisture levels. b): SHAP spatial distribution for the SHAP peak-day (4 August 2018) of *g500* lag 1. Statistical significance was assessed via a bootstrapping procedure. The 95% confidence interval for the mean SHAP value was determined by using the 2.5th and 97.5th percentiles of the resulting bootstrapped distribution. c): standardized anomalies of geopotential at 500hPa in colour and data in contours for the geopotential height at 500hPa during the SHAP peak day. The black dot in panel *b* marks the location of Hannover.

## 3.3 Case study: Summer heatwave Córdoba 2021

For the first case study, we test the CombinedModel during the period 10 August 2021 to 16 August 2021 in the Córdoba region, when a record-shattering heatwave struck the Iberian Peninsula. During this period, temperatures up to 47.6°C were registered in La Rambla (Córdoba) (AEMET, 2021). The model predicts correctly all the daily labels during this period: [0,1,1,1,1,1,1], with 0 corresponding to non-extreme class and 1 to extreme class.

Fig. 7a shows the temporal evolution of the extreme class SHAP values for the atmospheric circulation features in terms of mean absolute SHAP (spatially aggregated), and the three soil moisture levels. Fig. 7b represents the SHAP values for the *g500* feature for the SHAP peak in fig. 7a, with statistical significance assessed via a bootstrapping procedure using the 95% confidence interval determined with the 2.5th and 97.5th percentiles. The values outside the confidence interval are not plotted. Lastly, Fig. 7c displays the standardized anomalies for the *g500* field (colours) and the geopotential height at 500hPa data (black contours) during the peak day. During the thirteenth of August, the positive anomalies and anticyclonic pattern are right above the Iberian Peninsula, and our model gives mainly positive SHAP values for that region, showing strong locality around the predicted location. This signal starts to be strong and positive when the positive anomalies reach the predicted location (Fig. 7a).




The soil moisture anomalies during this event are negative but not especially low (not shown), resulting in SHAP values for the soil moisture levels being small and remaining quite constant. A curious result is the CO<sub>2</sub> being generally the second most important feature. This suggests that even for a specific extreme event like this one, we can identify a clear anthropogenic contribution.

#### 3.4 Case study: Summer heatwave Hannover 2018

We select the Hannover region for the second case study to have a contrasting soil-moisture regime compared to the Córdoba region. We focus on the 2018 European heatwave, which was an exceptional climatic event that brought prolonged periods of extreme heat and drought to much of Northern and Central Europe, including Germany.

The period selected is 29 July 2018 to 4 August 2018 and the model predicts correctly all classes, which are a complete week of temperatures above the 90th percentile: [1,1,1,1,1,1]. In Fig. 8 we show the same type of plots as in the former case study, with the same significance treatment in 8b as in the previous case. Again we see a strong locality in the spatial SHAP values for g500, with positive values surrounding the predicted location, where the positive anomalies are located. The spatial distributions are noisy for the rest of circulation variables, but they all explain less than 10% of the total mean absolute SHAP value. In Fig. 8a we can see that in this case the level one of soil moisture plays a more important role compared to the Córdoba event. The soil moisture at level one has positive SHAP values during the whole week, which comparable to g500 in magnitude during the first days. Regarding the CO<sub>2</sub> concentrations, in this case its relevance is much lower compared to the Córdoba case. Different drivers have been proven to play a central role in modulating the occurrence and intensity of summer temperature extremes, yet their individual contributions remain difficult to quantify. In this study, we develop an explainable machine-learning framework to disentangle the respective influences of large-scale atmospheric circulation, soil-moisture anomalies, and rising CO<sub>2</sub> concentrations on boreal-summer temperature extremes at six locations across Europe and North Africa with different characteristics of land-atmosphere coupling (Córdoba, Lyon, Hannover, Stockholm, Belgrade, and Marrakech). Using SHapley Additive exPlanation (SHAP) values, we find that the atmospheric circulation consistently dominates model explainability across all locations, contributing to 67-90 % of the total mean SHAP, with the geopotential at 500 hPa field contributing the most. Soil-moisture influence exhibits a northward gradient: negligible at Marrakech (0.5 %), moderate at Córdoba (7.7 %), and substantial at Lyon (15 %). Additionally, negative correlations between soil-moisture standardized anomalies and SHAP values across three depth levels corroborate the amplifying effect of land drying on heat extremes. We demonstrate the robustness of these findings to a less stringent (80th percentile) extreme definition. Furthermore, the identified driver contributions are consistent when using alternative observational data for temperature extreme definition and for computing SPI/SPEI drought indices as proxies for soil moisture, with SPEI showing a closer alignment to the original ERA5-Land results. We also illustrate the methodology for case studies of two individual events, heatwaves occurring in Córdoba (Spain) 2021 and Hannover (Germany) 2018, which reveal a pronounced spatial pattern in the distribution of SHAP values for the circulation predictors. They also confirm the enhanced role of the land component in regions of northern Europe and reveal a contribution of the anthropogenic factor through CO<sub>2</sub> concentrations, even for specific events. These insights enhance our understanding of the physical mechanisms behind temperature extremes and demonstrate the potential of explainable artificial intelligence methods to improve







the representation of land-atmosphere feedbacks in climate models. Different drivers have been proven to play a central role in modulating the occurrence and intensity of summer temperature extremes, yet their individual contributions remain difficult to quantify. In this study, we develop an explainable machine-learning framework to disentangle the respective influences of large-scale atmospheric circulation, soil-moisture anomalies, and rising CO2 concentrations on boreal-summer temperature extremes at six locations across Europe and North Africa with different characteristics of land-atmosphere coupling (Córdoba, Lyon, Hannover, Stockholm, Belgrade, and Marrakech). Using SHapley Additive exPlanation (SHAP) values, we find that the atmospheric circulation consistently dominates model explainability across all locations, contributing to 67-90 % of the total mean SHAP, with the geopotential at 500 hPa field contributing the most. Soil-moisture influence exhibits a northward gradient: negligible at Marrakech (0.5 %), moderate at Córdoba (7.7 %), and substantial at Lyon (15 %). Additionally, negative correlations between soil-moisture standardized anomalies and SHAP values across three depth levels corroborate the amplifying effect of land drying on heat extremes. We demonstrate the robustness of these findings to a less stringent (80th percentile) extreme definition. Furthermore, the identified driver contributions are consistent when using alternative observational data for temperature extreme definition and for computing SPI/SPEI drought indices as proxies for soil moisture, with SPEI showing a closer alignment to the original ERA5-Land results. We also illustrate the methodology for case studies of two individual events, heatwaves occurring in Córdoba (Spain) 2021 and Hannover (Germany) 2018, which reveal a pronounced spatial pattern in the distribution of SHAP values for the circulation predictors. They also confirm the enhanced role of the land component in regions of northern Europe and reveal a contribution of the anthropogenic factor through CO<sub>2</sub> concentrations, even for specific events. These insights enhance our understanding of the physical mechanisms behind temperature extremes and demonstrate the potential of explainable artificial intelligence methods to improve the representation of land-atmosphere feedbacks in climate models.

#### 4 Conclusions

In this study, we quantify the contribution of soil moisture, atmospheric circulation and CO<sub>2</sub> forcing to extreme temperature events during the boreal summer using an explainable ML approach. We focus on six locations with different land–atmosphere coupling regimes. Based on a percentile definition of extremes, we find an increasing trend in the yearly count of extreme days during summer for Córdoba, Lyon, Belgrade, and Marrakech regions.

To ensure the robustness of the model predictions, we train an ensemble of 20 members and analyse the 50% most confident predictions across all observed cases. Balanced accuracy increases consistently with the percentage of confidence using the CombinedModel, indicating stable predictive skill. The area under the ROC curve also improves with confidence. We further tested this robustness by applying our framework to different extreme definitions and data sources. The model trained on the 80th percentile definition showed equally robust predictive performance, while the model trained on the more stringent 95th percentile definition exhibited less stable behaviour due to the high class imbalance, leading us to disregard its explainability outputs. Furthermore, the framework demonstrated reliable performance when using observational data for TX and for com-




puting SPI/SPEI drought indices as proxys for soil moisture, with SPEI giving qualitatively more similar results to the model using ERA5 data for the atmospheric circulation and the soil moisture levels.

To interpret the predictions of the model, we use SHAP values, which allows us to assess the marginal contributions of the atmospheric circulation, land and atmopspheric  $CO_2$  features. The reader must keep in mind that the explainability results represent the interpretations done by the model, and we cannot assume that these results represent the reality. With the strategies explained in 2.7 we are ensuring a treatment of part of the existing epistemic uncertainty when using this model, but different architectures or different models might differ in the results. The explainability results show that circulation features, especially the geopotential at 500 hPa, dominate the prediction of extreme events across all locations (66.9–89.7 %). The land component shows a regional differing behaviour: its importance increases in temperate and northern regions (10.1–15.0 %), while remaining negligible or moderate in arid locations like Córdoba and Marrakech, representing 7.7 % and 0.5 % of the total SHAP respectively. The correlation between SHAP values and standardized soil moisture anomalies is negative across all levels, confirming the role of dry conditions in enhancing the probability of extreme heat events.  $CO_2$  SHAP contributions are substantial in locations where we detect a trend in the yearly extreme count, suggesting that the model captures the long-term forcing signal.

Furthermore, the core patterns of driver contributions identified by the model are robust to a different definition of the extreme events and to the use of observational data from E-OBS instead of ERA5-Land. The north–south gradient in land driver importance and the dominance of atmospheric circulation are consistently reproduced in the analysis using the 80th percentile extreme definition, albeit with a reduced relative contribution from soil moisture, likely due to the inclusion of less severe events. Moreover, the regionality of the land component is robustly recovered when using observational temperature data and the SPEI drought index as a proxy for soil moisture. The SPI index, which lacks the evapotranspiration component of SPEI, resulted in a weaker land signal, underscoring the importance of representing more accurately surface water balance for capturing land–atmosphere feedbacks and confirming that SPEI provides a more physically comprehensive proxy for this purpose.

We also analyse two case studies to further study the explainability technique. For the 2021 Córdoba heatwave, the model attributes the dominant contribution to the atmospheric circulation, with a mainly positive SHAP signal over the Iberian Peninsula when positive 500 hPa geopotential are placed in the region. We identify a clear anthropogenic contribution due to the high CO<sub>2</sub> SHAP value in this event. The land variables play a minor role in this case, with anomalies close to zero for the three soil moisture levels. During the 2018 Hannover heatwave, we see an enhanced importance of the land component, especially the first soil moisture level, with positive SHAP values comparable to the *g500* circulation feature during the first days of the event. Again, we see a strong locality in the spatial distribution of SHAP values. In this second case study, CO<sub>2</sub> plays a minor role.

These results reveal the central role of atmospheric circulation in driving extreme temperature events and highlight the regionally dependent modulation by soil moisture. The explainability technique used allows us to attribute data-driven predictions to physical drivers, offering a diagnostic framework that can assess the contribution of individual components to extreme




events. We have demonstrated that this framework and its conclusions are robust to the choice of extreme event threshold (if the class imbalance is not too strong) and to the use of observational data instead of reanalysis data.

Our findings on the role of atmospheric circulation in driving daily temperature extremes support the findings of Uckan et al. (2024), who showed that the geopotential field is the most relevant feature over 59.9 % of global land at a 1-day scale and 67.9 % at a 7-day scale. We confirm the importance of the atmospheric circulation using a different ML architecture, adding robustness to this result. Additionally, using a ConvNeXt model for atmospheric fields allows us to identify the spatial regions to which the model attributes higher importance, further demonstrating the ability of XAI to reveal spatial patterns in weather and climate problems, as shown by Davenport and Diffenbaugh (2021) and Mayer and Barnes (2021) using LRP. SHAP values, however, remain relatively unexplored for such pattern identification.

The role of soil moisture in modulating the occurrence of extremes has been highlighted by Materia et al. (2022) and Seneviratne et al. (2006), showing that dry soils intensify and prolong heat extremes, while wet soils suppress heatwaves. Here, we support the modulating effect of this land–atmosphere coupling by analysing the correlation between soil moisture anomalies and corresponding SHAP values. Our work contributes to disentangling the components of this coupling and quantifying their roles using XAI. We further quantify the influence of soil moisture at three different depth levels and find that, on weather time scales, the level closest to the surface generally plays the largest role.

Finally, we include a quantification of the role of atmospheric CO<sub>2</sub>, which is assessed by Materia et al. (2025) at seasonal scales but is not usually considered in similar studies using explainable ML. Including this feature allows us to account for the trend in the occurrence of hot extremes due to global warming, without encountering the complications of detrending the data when splitting it into training, validation, and testing sets in ML studies.

Future work should include testing the robustness of the results across different model architectures. Analysing additional locations in future work may help gain insights into the regional variability of the results. Including variables like humidity fluxes or information on aerosol composition in each location could also provide complementary information on the physical processes modulating temperature extremes during boreal summer. The presented XAI methodology could also be applied to evaluate the importance of the different drivers in CMIP6 dynamical models or in Subseasonal to Seasonal to Decadal Prediction Systems.

Overall, this study provides a physically interpretable, data-driven framework to disentangle the contributions of atmospheric circulation, land surface conditions, and anthropogenic forcing to summer temperature extremes. By combining a robust machine learning method with an ML explainability technique, we offer insights that can inform both model development and climate attribution efforts. Our results underscore the dominant influence of atmospheric circulation and the modulating role of soil moisture, revealing consistent regional signals across different locations. These findings highlight the potential of explainable machine learning as a complementary tool for climate diagnostics and pave the way for more comprehensive, process-based assessments of future temperature extremes.

Code and data availability. The repository with the code used in this work is available in https://gitlab.earth.bsc.es/agarci8/quantifydrivershw. git. The repository contains the scripts needed to pre-process data (anomalies, heatwave dection), the deep learning model code toghether with the scripts for training the model and computing the explainability results, and scirpts or notebooks to post-process the results.

Additionally, the repository includes a few datasets with: the file with the data for the  $CO_2$  feature, the files with the SPEI and SPI drought indices, and the files with the optimized values for the hyperparameters of the deep learning model.

## Appendix A



#### A1 LOESS smoothing

To improve the robustness of daily climatology percentiles estimates, we use a local polynomial regression (LOESS) as proposed in Mahlstein et al. (2015). This method smooths the raw climatology by fitting a regression locally at each point using a subset of neighbouring days weighted by distance. The weights  $w_i$  applied to each point in the local window are based on their normalized distance  $d_i$  to the estimation point, using the tricubic function:

$$w_i = \begin{cases} (1 - |d_i|^3)^3, & \text{if } |d_i| \le 1\\ 0, & \text{if } |d_i| > 1 \end{cases}$$

where  $d_i = \frac{x_i - x}{h}$  is the distance between the input point  $x_i$  and the target point x, scaled by the distance h to the farthest neighbour in the local window. This ensures that the weights smoothly decay to zero at the edges of the window and are strictly zero beyond it.

In practice, this means we apply LOESS to the raw daily climatology or percentile values to obtain a smoother and more stable annual profile. This is particularly important when dealing with small sample sizes or high daily variability, where simple averaging can lead to noisy or misleading estimates. This smoothing step ensures that the resulting threshold and climatology estimates are not overly influenced by isolated values or short-term fluctuations, which is especially relevant for extreme event detection on a daily basis.

## A2 Cross-entropy loss function

The Cross Entropy Loss function is defined as:

$$\mathcal{L} = \frac{-1}{N} \sum_{i} y_i \cdot log(p_i) + (1 - y_i) \cdot log(1 - p_i)$$
 (A1)

where  $p_i$  is the probability of class 1,  $(1 - p_i)$  the probability of class 0, and  $y_i$  is the true label. This loss function is widely used in binary classification tasks in ML.

## Appendix B: Statistical and robustness results

**Table B1. Summary of extreme event counts by site and period for different extreme definitions.** Yearly extreme count sum is shown for the different locations, both for the training (1950–2014) and test (2014–2024) periods. **E** corresponds to the extreme class and **NE** to the non-extreme class.

| Site      | Period            | Total Days | 80th Percentile |      | 90th Percentile |      | 95th Percentile |      |
|-----------|-------------------|------------|-----------------|------|-----------------|------|-----------------|------|
|           |                   |            | E               | NE   | E               | NE   | E               | NE   |
| Cordoba   | Train (1950–2014) | 5888       | 1440            | 4448 | 795             | 5093 | 434             | 5454 |
|           | Test (2014–2023)  | 920        | 452             | 468  | 316             | 604  | 208             | 712  |
| Lyon      | Train (1950–2014) | 5888       | 1377            | 4511 | 783             | 5105 | 454             | 5434 |
|           | Test (2014–2023)  | 920        | 430             | 490  | 312             | 608  | 228             | 692  |
| Hannover  | Train (1950–2014) | 5888       | 1316            | 4572 | 702             | 5186 | 377             | 5511 |
|           | Test (2014–2023)  | 920        | 329             | 591  | 200             | 720  | 136             | 784  |
| Stockholm | Train (1950–2014) | 5888       | 1295            | 4593 | 647             | 5241 | 333             | 5555 |
|           | Test (2014–2023)  | 920        | 291             | 629  | 167             | 753  | 114             | 806  |
| Belgrado  | Train (1950–2014) | 5888       | 1368            | 4520 | 739             | 5149 | 397             | 5491 |
|           | Test (2014–2023)  | 920        | 323             | 597  | 200             | 720  | 122             | 798  |
| Marrakech | Train (1950–2014) | 5888       | 1339            | 4549 | 728             | 5160 | 386             | 5502 |
|           | Test (2014–2023)  | 920        | 316             | 604  | 194             | 726  | 120             | 800  |

Figure B1. Change in accuracies for the Combined Model for different percentages of confidence using the 90th percentile extreme definition. In the x-axis the percentage of confidence taken, and in the y-axis the accuracies. EA, NEA and BA in the legend stand for extreme accuracy, non-extreme accuracy and balanced accuracy respectively.

Figure B2. Change in ROC curves for the Combined Model for different percentages of confidence using the 90th percentile extreme definition. In the x-axis the false positive rate and in the y-axis the true positive rate. The legend shows the different percentages of confidence taken.

Figure B3. Same plot as in Fig. B1 but for the 80th percentile definition of extreme.

Figure B4. Same plot as in Fig. B1 but for the 95th percentile definition of extreme.

# **Appendix C: Drought indices correlations**

Figure C1. Spearman correlation between the three soil moisture levels (swvl1, swvl2, swvl3) and the SPEI index computed at three different time scale using the E-OBS gridded data. In the x-axis the scale used to compute the SPEI index and in the y-axis the resulting spearman correlation. The legend indicates the three original soi moisture levels used in the DL model.

Figure C2. Same plot as Fig. C1 but using the SPI drought index instead.

# 480 Appendix D: XAI results

Figure D1. Same plot as in Fig. 5 but for the 80th percentile definition of extreme.

Figure D2. Same plot as in Fig. 5 but for the model using E-OBS observational data and the SPEI index as a proxy for the soil moisture feature. The 90th percentile is still used for the extreme event definition.

Figure D3. Same plot as in Fig. 6 but for the model using E-OBS observational data and the SPEI 30 days index as a proxy for the soil moisture feature. The 90th percentile is still used for the extreme event definition.

Figure D4. Same plot as in Fig. 5 but for the model using E-OBS observational data and the SPI 30 days index as a proxy for the soil moisture feature. The 90th percentile is still used for the extreme event definition.

Figure D5. Same plot as in Fig. 5 but for the model using only g500 in the atmospheric circulation features.

Author contributions. AG, LP and MD did the conceptualization of the study; AG and LP worked on the data and developed the model code and methodology; AG performed the simulations; AG, LP, MD, SM and RM analysed the results; AG wrote the manuscript draft; LP, MD, SM, and RM reviewed and edited the manuscript; MD, SM and RM supervised the study; MD did the funding acquisition

Competing interests. The authors declare that they have no conflict of interest.

Acknowledgements. This research was conducted within the framework of the EXPECT project. EXPECT has received funding from the European Union's Horizon Europe Framework Programme (HORIZON) under Grant Agreement 101137656. Stefano Materia also acknowledges also acknowledges AI4S fellowships within the "Generación D" initiative by Red.es, Ministerio para la Transformación Digital y de la Función Pública of Spain, for talent attraction (C005/24-ED CV1), funded by NextGenerationEU through PRTR. The authors also want to thank Vincent Verjans for reviewing the manuscript.

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
