# Peer review of "Quantifying atmospheric and land drivers of hot temperature extremes through explainable Artificial Intelligence"

_EGUsphere, 2025_

## Referee Comment (RC2)

**Review of "*Quantifying atmospheric and land drivers of hot temperature extremes through explainable Artificial Intelligence*"**

The authors present an explainable machine learning framework to disentangle contributions of atmospheric circulation, soil moisture, and $CO_2$ to summer temperature extremes across six European/North African locations. The methodological approach is interesting and the paper is generally well-structured, but I have concerns regarding some of the interpretations and the framing of novelty. I also note that my expertise does not extend to the technical aspects of SHAP, so my comments focus primarily on the climate science dimensions.

**Main comments**

The heterogeneity in $CO_2$ SHAP contributions across locations (Fig. 5) is puzzling. If $CO_2$ is capturing the anthropogenic warming signal, one would expect relatively homogeneous contributions, perhaps modulated by maritime/continental influences. Instead, values range from 5.6% (Hannover) to 21.8% (Córdoba), and this pattern does not obviously map onto any physical explanation. I suspect $CO_2$ may be acting as a proxy for the location-specific trend in extreme frequency (cf. Fig. 4) rather than cleanly representing anthropogenic forcing. This warrants discussion and suggests non-negligible methodological uncertainty.

Related to this: given the 20-member ensemble, it should be straightforward to show uncertainties on the SHAP values. This is notably absent and would substantially strengthen confidence in the reported differences between locations and features.

The claims regarding physical understanding (e.g., L. 352: "*These insights enhance our understanding of the physical mechanisms behind temperature extremes*") should be tempered. The role of atmospheric circulation and land–atmosphere feedbacks in modulating heat extremes is well-established from both observations (e.g., Hirschi et al., 2011) and model experiments spanning at least two decades (e.g., Schär et al., 1999; Koster et al., 2004; Fischer et al., 2007; Wehrli et al., 2019; Wehrli et al., 2022). The contribution here is methodological (demonstrating the applicability of XAI techniques to this problem) rather than providing fundamentally new physical insight. The paper would benefit from framing it as such, and from engaging more thoroughly with existing literature (of which I have only cited some examples).

On a related note, some claims would be more compelling if results were shown primarily for "*pure*" observations. Since ERA5-Land is an offline land surface model simulation, its behaviour bears similarity to land components of ESMs that have long pointed to these feedbacks. The robustness tests using E-OBS and SPEI are appreciated, but greater emphasis on observational constraints would strengthen the analysis.

**Specific comments**

**L. 69:** The term "*arid*" is imprecise here. Hsu & Dirmeyer (2023) distinguish "*dry,*" "*transitional,*" and "*wet*" soil moisture–evaporation regimes; Córdoba would fall into a dry/transitional regime rather than being truly arid climatologically.

**L. 80:** Hersbach et al. (2020) should be cited for ERA5.

**L. 87 vs. L. 116:** The justification for using ERA5-Land based on resolution is unconvincing given that the analysis ultimately operates at 1° resolution with 100 km spatial averaging. The more relevant advantage is that ERA5-Land, as an offline (land surface model) simulation, does not assimilate screen-level observations to adjust soil moisture (unlike ERA5), meaning soil moisture variability is more physically consistent.

**Section 2.2:** It would help to briefly explain upfront why SPI/SPEI are introduced, since the main analysis relies heavily on ERA5-Land and the drought indices only appear much later.

**L. 165:** The statement that "*the number of extreme days... remains low*" is confusing given that ~10% of days are extreme by construction. The intended meaning (class imbalance challenges for ML) should be clarified.

**L. 223:** The claim that Belgrade shows "*a visible trend after approximately 1960*" likely applies to other locations as well. It would be useful to conduct formal trend analysis with significance testing for different periods (e.g., full record vs. 1980 onwards), particularly since continental locations may have experienced aerosol-related dimming effects in earlier decades.

**L. 229:** "*table 1 in ??*"

**L. 77:** "*sect. ??*"

**L. 336, 354:** There appears to be duplicated/misplaced text here (the abstract seems to be repeated). Please check.

**Fig. 5:** See major comment above regarding $CO_2$ heterogeneity.

**Fig. 6a:** It is encouraging to see that deeper soil layers contribute more in Córdoba, consistent with top soils being fully desiccated in summer. This physical interpretation could be mentioned explicitly.

**Figs. 7 & 8:** Showing temperature anomalies (e.g., as background shading) would help readers understand when the temperature peak occurs relative to the SHAP evolution—particularly for the Hannover case where g500 SHAP keeps increasing rather than weakening after what one might assume is the peak. Also: "*standarized*" is a typo in panel c, and the colour schemes differ between figures without obvious reason.

**L. 383:** Given the pattern of increasing land component importance from the 80th to 90th percentile, it might be worth acknowledging that land variables could play an even more important role for more extreme events (even if the 95th percentile analysis is not robust).

**L. 434:** The finding regarding the first soil moisture level being most important should be qualified as applying to humid/transitional regions, since it clearly does not hold for drier locations like Córdoba (cf. Fig. 6a).

**References**

Hersbach, H., et al. (2020). The ERA5 global reanalysis. *Q. J. R. Meteorol. Soc.*, 146, 1999–2049.

Hirschi, M., et al. (2011). Observational evidence for soil-moisture impact on hot extremes in southeastern Europe. *Nat. Geosci.*, 4, 17–21.

Hsu, H., & Dirmeyer, P. A. (2023). Soil moisture-evapotranspiration coupling shifts into new gears under increasing $CO_2$. *Nat. Commun.*, 14, 1162.

Fischer, E. M., Seneviratne, S. I., Vidale, P. L., Lüthi, D., & Schär, C. (2007). Soil moisture–atmosphere interactions during the 2003 European summer heat wave. *J. Clim.*, 20, 5081–5099. https://doi.org/10.1175/JCLI4288.1

Koster, R. D., et al. (2004). Regions of strong coupling between soil moisture and precipitation. *Science*, 305, 1138–1140.

Schär, C., Lüthi, D., Beyerle, U., & Heise, E. (1999). The soil-precipitation feedback: A process study with a regional climate model. *J. Clim.*, 12, 722–741.

Wehrli, K., Guillod, B. P., Hauser, M., Leclair, M., & Seneviratne, S. I. (2019). Identifying key driving processes of major recent heat waves. *J. Geophys. Res. Atmos.*, 124, 11746–11765. https://doi.org/10.1029/2019JD030635

Wehrli, K., Luo, F., Hauser, M., Shiogama, H., Tokuda, D., Kim, H., Coumou, D., May, W., Le Sager, P., Selten, F., Martius, O., Vautard, R., & Seneviratne, S. I. (2022). The ExtremeX global climate model experiment: investigating thermodynamic and dynamic processes contributing to weather and climate extremes. *Earth Syst. Dynam.*, 13, 1167–1196. https://doi.org/10.5194/esd-13-1167-2022